

# Next-generation sequencing of *BRCA1* and *BRCA2* genes for rapid detection of germline mutations in hereditary breast/ovarian cancer

Arianna Nicolussi[1], Francesca Belardinilli[2], Yasaman Mahdavian[2], Valeria Colicchia[2], Sonia D'Inzeo[1,3], Marialaura Petroni[4], Massimo Zani[2], Sergio Ferraro[2], Virginia Valentini[2], Laura Ottini[2], Giuseppe Giannini[2,5], Carlo Capalbo[2] and Anna Coppa[1]

[1] Department of Experimental Medicine, University of Roma "La Sapienza", Roma, Italy
[2] Department of Molecular Medicine, University of Roma "La Sapienza", Roma, Italy
[3] U.O.C. Microbiology and Virology Laboratory, A.O. San Camillo Forlanini, Roma, Italy
[4] Istituto Italiano di Tecnologia, Center for Life Nano Science@Sapienza, Roma, Italy
[5] Istituto Pasteur-Fondazione Cenci Bolognetti, Roma, Italy

Corresponding authors
Giuseppe Giannini,
giuseppe.giannini@uniroma1.it
Anna Coppa,
anna.coppa@uniroma1.it

## ABSTRACT

**Background:** Conventional methods used to identify *BRCA1* and *BRCA2* germline mutations in hereditary cancers, such as Sanger sequencing/multiplex ligation-dependent probe amplification (MLPA), are time-consuming and expensive, due to the large size of the genes. The recent introduction of next-generation sequencing (NGS) benchtop platforms offered a powerful alternative for mutation detection, dramatically improving the speed and the efficiency of DNA testing. Here we tested the performance of the Ion Torrent PGM platform with the Ion AmpliSeq BRCA1 and BRCA2 Panel in our clinical routine of breast/ovarian hereditary cancer syndrome assessment.

**Methods:** We first tested the NGS approach in a cohort of 11 patients (training set) who had previously undergone genetic diagnosis in our laboratory by conventional methods. Then, we applied the optimized pipeline to the consecutive cohort of 136 uncharacterized probands (validation set).

**Results:** By minimal adjustments in the analytical pipeline of Torrent Suite Software we obtained a 100% concordance with Sanger results regarding the identification of single nucleotide alterations, insertions, and deletions with the exception of three large genomic rearrangements (LGRs) contained in the training set. The optimized pipeline applied to the validation set (VS), identified pathogenic and polymorphic variants, including a novel *BRCA2* pathogenic variant at exon 3, 100% of which were confirmed by Sanger in their correct zygosity status. To identify LGRs, all negative samples of the VS were subjected to MLPA analysis.

**Discussion:** Our experience strongly supports that the Ion Torrent PGM technology in *BRCA1* and *BRCA2* germline variant identification, combined with MLPA analysis, is highly sensitive, easy to use, faster, and cheaper than traditional (Sanger sequencing/MLPA) approaches.

## INTRODUCTION

Inherited mutations in highly penetrant *BRCA1* and *BRCA2* genes occur in about 5–10% of breast and ovarian cancer disease (*Foulkes, 2008*). A recent prospective cohort study has estimated that the cumulative risks of breast cancer to age 80 years was 72% for *BRCA1* and 69% for *BRCA2* carriers (*Kuchenbaecker et al., 2017*). A large mutation spectrum has been reported for these genes, as recently revised (*Rebbeck et al., 2018*). The contribution of low-penetrance and risk-modifying genetic polymorphisms to a more appropriate assessment of individual risk is also emerging (*Couch et al., 2012*; *Ottini et al., 2013*; *Kuchenbaecker et al., 2014*; *Peterlongo et al., 2015*).

Direct Sanger sequencing represented the most common method for over two decades to identify single nucleotide alterations, insertions, and deletions of *BRCA1* and *BRCA2* in the clinical practice. Large genomic alterations may be detected by the multiplex ligation-dependent probe amplification (MLPA) assay. Sanger sequencing and MLPA are to date considered the gold standard methods to determine the *BRCA1* and *BRCA2* mutation status. However, the big size (5,592 bp and 10,257 bp, respectively), the content in homopolymeric regions and the high-allelic heterogeneity of the genes, together with the lack of mutation hot spots, made the diagnostic procedure time-consuming and costly. The recent improvements in DNA sequencing technology, and the emergence of benchtop next-generation sequencing (NGS) instruments offered a powerful alternative for mutation detection, dramatically improving the speed and the efficiency of DNA testing (*Feliubadaló et al., 2013*; *Yeo et al., 2014*; *Trujillano et al., 2015*). Indeed, NGS significantly speeded the reporting times of hereditary breast/ovarian cancer predisposition, important for rapid clinical management of carriers.

The implementation of targeted therapy in solid cancers with poly(ADP-ribose) polymerase (PARP) inhibitors (*O'Sullivan Coyne, Chen & Kummar, 2015*; *Colicchia et al., 2017*; *Stewart, Pilié & Yap, 2018*) and in particular the observation of their effectiveness in some phase 3 trials carried out in women with *BRCA1* or *BRCA2* pathogenic variants and metastatic ovarian or advanced breast cancer (*Coleman et al., 2017*; *Robson et al., 2017*; *Oza et al., 2018*; *Litton et al., 2018*), indicate *BRCA1* and *BRCA2* mutation status as a novel biomarker and highlights the importance of shortening the turn-around time (TAT) of the analysis to give results and to make possible a faster identification of carriers. To respond to the increasing request of genetic testing, NGS methods are starting to be used routinely in many molecular genetic laboratories including ours.

Next-generation sequencing technology is currently used with several available platforms, such as the Genome Sequencer (Roche-454 Life Sciences, Indianapolis, IN, USA), the Genome Analyzer/HiSeq/MiSeq (Illumina-Solexa, San Diego, CA, USA), Ion Torrent PGM, and Ion Proton sequencers (Thermo Fisher Scientific, Waltham, MA, USA), and the HeliScope from Helicos BioSciences (Cambridge, MA, USA) (*Voelkerding, Dames & Durtschi, 2009*; *Rothberg et al., 2011*; *Wallace, 2016*). Key differences exist in their

performance, quality of data, and throughput capacity: MiSeq had the highest throughput per run (1.6 Gb/run, 60 Mb/h) and lowest error rates, 454 GS Junior generated the longest reads (up to 600 bases) but had the lowest throughput (70 Mb/run, nine Mb/h), while Ion Torrent PGM had the highest throughput (80–100 Mb/h) (*Loman et al., 2012*). Ion Torrent PGM is a new sequencing platform that substantially differs from other sequencing technologies by measuring pH rather than light to detect polymerization events. It represents the first "post-light" sequencing instrument that exploits the emulsion PCR and incorporates a sequencing-by-synthesis approach. All four nucleotides are provided in a stepwise fashion during an automated run. When the nucleotides, complementary to the template are incorporated into the nascent strand, the hydrolysis of the incoming nucleotide triphosphate causes the net liberation of a single proton for each nucleotide incorporated, producing a shift in the pH of the surrounding solution detected by a modified silicon chip (*Rothberg et al., 2011*). Ion Torrent PGM offers three differently priced sequencing-chip reagents, giving a large flexibility in designing experiments, as a choice can be made based on the throughput required. In the present study, we evaluated the performance of the Ion Torrent PGM platform in combination with the ready-to-use Ion AmpliSeq BRCA1 and BRCA2 Panel (Life Technologies, Carlsbad, CA, USA) in the diagnostic screening for the identification of single nucleotide alterations, insertions, and deletions in *BRCA1* and *BRCA2* genes.

## MATERIALS AND METHODS

### Patients and DNA

A retrospective cohort of 11 familial breast cancer patients, training set (TS), previously tested by Sanger sequencing for all *BRCA1* and *BRCA2* exons and by MLPA to detect large genomic rearrangements (LGRs), and a prospective consecutive cohort of 136 uncharacterized probands, validation set (VS), enrolled at the Hereditary Tumors section of Policlinico Umberto I, University La Sapienza, between July 2015 and September 2017, were analyzed. Selection criteria were as follows: (i) three or more breast cancer cases diagnosed at any age or two first-degree family members affected before 50; (ii) early onset breast cancer (<35 years); (iii) breast and ovarian cancer in the same individual or two breast cancer cases and at least one ovarian cancer case, or one breast cancer case and one ovarian cancer case diagnosed before 50 in first-degree family members; (iv) male breast cancer; (v) BRCAPro5 greater than 10% (*Capalbo et al., 2006a*, *2006b*; *Coppa et al., 2014*).

Comprehensive pre-test counseling was offered to all probands and their family members and informed consent were obtained. The TS was characterized by 11 samples carrying 11 different pathogenic mutations including single nucleotide variants (SNVs), indels, and LGRs plus 37 benign variants (Table 1; Table S1).

Genomic DNA was extracted from peripheral blood samples using a commercial kit (QIAamp Blood Kit, Qiagen, Valencia, CA, USA) according to the manufacturer's instructions. All investigations were approved by Ethics Committee of the University of Roma "La Sapienza" (Prot.: 88/18; RIF.CE:4903, 31-01-2018) and conducted according to the principles outlined in the declaration of Helsinki.

**Table 1  BRCA1 and BRCA2 variants contained in the training set used for the optimization of the workflow.**

| Variant | | Gene | Effect | Type | Frequency (%) | Phred quality score | Coverage | Coverage var |
|---|---|---|---|---|---|---|---|---|
| **DNA level** | **Protein level** | | | | | | | |
| c.5266dupC | p.Gln1756Profs | BRCA1 | Pathogenic | dup | 50.7 | 4,616 | 2,362 | 1,198 |
| c.1504_1508delTTAAA | p.Leu502Alafs | BRCA1 | Pathogenic | del | 481 | 1,492.07 | 778 | 374 |
| c.4964_4982del19 | p.Ser1655Tyrfs | BRCA1 | Pathogenic | del | 39.6 | 50 | 844 | 334 |
| c.5407-?_*(1_?)del | p.? | BRCA1 | Pathogenic | exon 23-24del | / | / | / | / |
| c.5075-?_5193+?del | p.? | BRCA1 | Pathogenic | exon 18-19del | / | / | / | / |
| NBR2delEX1_BRCA1delEX1-2 | p.? | NBR2/BRCA1 | Pathogenic | exon 1del (NBR2) exon 1-2del (BRCA1) | / | / | / | / |
| c.7921_7926delGAATTTinsAG | p.Glu2641Argfs | BRCA2 | Pathogenic | complex | 44.6 | 1,094.44 | 673 | 300 |
| c.246dupA | p.Glu83Argfs | BRCA2 | Pathogenic | dup | 44.7 | 409.21 | 253 | 113 |
| c.2806_2809delAAAC | p.Ala938Profs | BRCA2 | Pathogenic | del | 48.2 | 1,144.45 | 591 | 285 |
| c.516+1G>C | p.? | BRCA2 | Pathogenic | SNV | 49.3 | 4,602.4 | 2,286 | 1,126 |
| c.7007G>A | p.Arg2336His | BRCA2 | Pathogenic | SNV | 48.4 | 1,128.61 | 576 | 279 |

Note:
Abbreviations: del, deletion; dup, duplication; SNV, single nucleotide variant; Phred quality score: probability of incorrect call.

## Ion Torrent PGM library preparation

The entire coding regions of the *BRCA1* and *BRCA2* genes were amplified using the Ion AmpliSeq BRCA1 and BRCA2 Panel (Life Technologies, Carlsbad, CA, USA) consisting of three primer pools, covering the target regions in 167 amplicons, including all exons and 10–20 bp of intronic flanking sequences, for both genes. Ion Torrent adapter-barcode ligated libraries were generated using Ion AmpliSeq™ Library Kit 2.0 (Applied Biosystems, Houston, TX, USA; Life Technologies, Carlsbad, CA, USA) and Ion Xpress™ Barcode Adapter 1-16 Kit (Applied Biosystems, Houston, TX, USA; Life Technologies, Carlsbad, CA, USA) according to the manufacturer's procedures to obtain 200-bp PCR fragments flanked by adaptor and barcode sequences, allowing sequencing and sample identification, respectively. In brief, DNA concentration was measured with Qubit™ dsDNA HS Assay Kit (Thermo Fisher Scientific, Waltham, MA, USA) and Qubit® Fluorometer (Thermo Fisher Scientific, Waltham, MA, USA). Ten ng of each genomic DNA sample was amplified, using Ion AmpliSeq™ HiFi Master Mix (Applied Biosystems, Houston, TX, USA; Life Technologies, Carlsbad, CA, USA) and Ion AmpliSeq *BRCA1* and *BRCA2* Community Panel primer pools (3 pools), for 2 min at 99 °C, followed by 19 two-step cycles of 99 °C for 15 s and 60 °C for 4 min, ending with a holding period at 10 °C in a GeneAmp PCR System 9700 thermocycler (Applied Biosystems, Houston, TX, USA). The amplified DNA samples were digested, then barcodes and adapters were ligated to the amplicons as previously described (*Belardinilli et al., 2015*) and reported in Ion AmpliSeq library preparation manual. Adaptor ligated amplicon libraries were purified with the Agencourt AMPure XP system (Beckman Coulter Genomics, Danvers, MA, USA). Following Agencourt AMPure XP purification, the concentration of the libraries was determined using the Qubit™ dsDNA HS Assay Kit

(Thermo Fisher Scientific, Waltham, MA, USA) and Qubit® Fluorometer (Thermo Fisher Scientific, Waltham, MA, USA). After quantification, each amplicon library was diluted to 50 pM and the same amount of the 12 libraries for four patients (Chip 314v2) or 24 libraries for eight patients (Chip 316v2) was pooled to perform the emulsion PCR reaction and the sequence reaction.

## Emulsion PCR and sequencing

The emulsion PCR was carried out with the Ion OneTouch 2 System and Ion Torrent PGM Hi-Q OT2 Kit (Thermo Fisher Scientific, Waltham, MA, USA) according to the manufacturer's procedure. Template-positive Ion Sphere™ Particles were enriched with the Dynabeads MyOne Streptavidin C1 Beads (Thermo Fisher Scientific, Waltham, MA, USA) and washed with Ion OneTouch Wash Solution. This process was performed using an Ion OneTouch ES system (Thermo Fisher Scientific, Waltham, MA, USA). After the Ion Sphere Particle preparation, Massive Parallel Sequencing was carried out with Ion Torrent PGM sequencer system (Thermo Fisher Scientific, Waltham, MA, USA) using the Ion Torrent PGM Hi-Q Sequencing Kit and Ion 314v2 or 316v2 Chip (Thermo Fisher Scientific, Waltham, MA, USA), according to the established procedures.

## Data analysis

Raw sequence data analysis, including base calling, demultiplexing, alignment to the hg19 human reference genome (Genome Reference Consortium GRCh37), was performed using the Torrent Suite Software version 3.6 and subsequent versions up to 5.2 (Thermo Fisher Scientific, Waltham, MA, USA). The average depth of total coverage was set at minimum $500\times$ and for variant calls at minimum of $100\times$.

The TS data were processed using Coverage Analysis and Variant Caller (germline low stringency) plugins available within the Torrent Suite Software version 3.6 using default analysis settings and subsequently using the version 3.6 of json parameters file associated with the *BRCA1* and *BRCA2* Ion Ampliseq Panel. The VS data were processed using Coverage Analysis and Variant Caller plugins available within the Torrent Suite Software versions from 4.0 to 5.2 and the json parameters file associated with the *BRCA1* and *BRCA2* Ion Ampliseq Panel.

All the variants were visually examined and verified using integrative genomics viewer (IGV) version 5.01 (http://www.broadinstitute.org/igv) as well as by filtering out possible strand-specific errors (i.e. mutations detected exclusively in the "plus" or in the "minus" strand, but not in both DNA strands). All mutations are reported following the human genome variation society guidelines (http://www.hgvs.org/mutnomen/) on the basis of the coding sequences NM_007294.3 and NM_000059.3 for *BRCA1* and *BRCA2*, respectively. Common polymorphisms, which represent 5% in the general population, and variant of uncertain significance (VUS) or pathogenic variants were classified referring to the following databases: Breast Cancer Information Core BIC (https://research.nhgri.nih.gov/), Clinical Variants (https://www.ncbi.nlm.nih.gov/pubmed), dbSNP138 (https://www.ncbi.nlm.nih.gov/projects/SNP/), ARUP (http://arup.utah.edu/database/BRCA/),

Universal Mutation Database (http://www.umd.be/BRCA1/, http://www.umd.be/BRCA2/), Leiden Open (source) Variation Database (LOVD) (http://www.lovd.nl/3.0/home). Unclassified variants were evaluated by the following *in silico* predictors: SIFT-PROVEAN (http://provean.jcvi.org/index.php), Polyphen (http://genetics.bwh.harvard.edu/pph2/), GVGD (http://agvgd.hci.utah.edu/agvgd_input.php), Mutation taster (http://www.mutationtaster.org/), Fruit Fly Splice Predictor, NNSPLICE (http://www.fruitfly.org/seq_tools/splice.html), Human Splicing Finder (http://www.umd.be/HSF3/), Splice Predictor (DK), NetGene2 (http://www.cbs.dtu.dk/services/NetGene2/), MaxEntScan (http://genes.mit.edu/burgelab/maxent/Xmaxentscan_scoreseq.html), ESEfinder3.0 (http://rulai.cshl.edu/cgi-bin/tools/ESE3/esefinder.cgi?process=home). The meaning of the variants not yet reported in the literature and in the databases was established as described by *Giannini et al. (2008)*, according to ENIGMA criteria (https://enigmaconsortium.org/).

Concordance of the NGS results with respect to those obtained with Sanger sequencing was calculated using classifications as true positive (TP), true negative (TN), false negative (FN), or false positive (FP). Regarding specificity and negative predictive value (NPV), since these parameters are calculated considering TNs, we considered the number of wild-type bases that were called by NGS and aligned to the reference sequence.

## Sanger sequencing

A subset of clinical samples (11 samples of TS and 60 samples of VS) were sequenced for the entire coding regions by Sanger sequencing. Sequencing was performed using an ABI PRISM DyeDeoxy Terminator Cycle Sequencing Kit and an ABI 3100 Genetic Analyzer (Applied Biosystems, Warrington, UK) according to *Coppa et al. (2018)*.

The reference sequence for *BRCA1* was NM_007294.3, and the reference sequence for *BRCA2* was NM_000059.3.

## MLPA assay

*BRCA1* and *BRCA2* genomic rearrangements were searched by the MLPA methodology (*Schouten et al., 2002*) according to the manufacturer's instructions (MRC-Holland, Amsterdam, The Netherlands) and as described (*Buffone et al., 2007*; *Coppa et al., 2018*). Variations in peak height were evaluated comparing each sample with a normal control and by a cumulative comparison.

## RESULTS

### Coverage data analysis

The *BRCA1* and *BRCA2* panel used in this study generates 167 amplicons that cover all targeted coding exons and exon-intron boundaries. The data coverage in 147 patients (TS and VS) showed a mean amplicon reading depth per sample ranging from 244× to 2,236×, loading 2/4 samples on chips 314v2 or eight samples on chip 316v2. An adequate number of reads mapped onto target regions has been obtained both with chip 314v2 or chip 316v2, with a mean sequencing depth and an average uniformity of coverage of 726× and 97% and 1,504× and 97%, respectively (Table 2; Table S2 and S3). The large increase in sequencing efficiency observed with the 316v2 chips makes the analysis of a

**Table 2** Mean quality control and coverage metrics in the training set and in the validation set samples.

| | Chip 314 ($n = 84$) | | Chip 316 ($n = 63$) | |
|---|---|---|---|---|
| | Mean | SD | Mean | SD |
| Number of mapped reads | 122,109 | 43,431 | 244,737 | 69,845 |
| Percent reads on target (%) | 96 | 0.03 | 97 | 0.02 |
| Uniformity of base coverage (%) | 97 | 0.02 | 97 | 0.01 |
| Average base coverage depth | 726 | 266 | 1,504 | 379 |

better quality and led to a lower number of false-positive indel calls. For variant calls, we chose the threshold of a minimum average depth of $100\times$, although a minimum average of $80\times$ is normally required for germline mutations (*Trujillano et al., 2015*) and a minimum average depth in strand bias of $30\times$ for each strand end to end. By a critical analysis of the *BRCA1* and *BRCA2* coverage data performed on 147 samples (TS and VS) emerged that some amplicons (herein defined as lower-performance amplicons) often did not reach a minimum coverage of $100\times$ or showed fwd/rev end to end unbalance (Table 3; Fig. 1). In particular, the coverage of the AMPL225438570 *BRCA1* amplicon was <$100\times$ in 61% of samples (Fig. 1A). This amplicon maps in the exon 2 of the *BRCA1* gene, containing a large A-T reach region, likely to be responsible for lower amplification efficiency. Concerning *BRCA2*, we observed five amplicons with a coverage less than $100\times$, whose GC content, self-annealing/hairpin formation and GC clamp issues could be responsible for low performance (Table 3; Fig. 1B). The same issues may also be responsible for fwd/rev end to end unbalance observed in the following amplicons: AMPL223735053, AMPL223413081, AMPL223712774 in *BRCA1* (Figs. 1C and 1D; Table 3); AMPL224626553, AMPL223938117, AMPL223512592, AMPL225349438, AMPL223730984, AMPL223515418, AMPL225441321, AMPL223959719 in *BRCA2* (Figs. 1E and 1F; Table 3). Of note, the read depths of these amplicons did not increase proportionally with an increase in mean amplicon read depths (across 167 amplicons) per sample. No variants have been identified in lower-performance amplicons in our TS and VS samples, using Ion AmpliSeq BRCA1 and BRCA2 Panel (Life Technologies, Carlsbad, CA, USA). However, all amplicons with strand bias were carefully and visually verified with IGV and re-analyzed by Sanger sequencing that confirmed the absence of variants.

## Detection of *BRCA1* and *BRCA2* variants

The TS, used to test the performance of the Ion AmpliSeq BRCA1 and BRCA2 Panel (Life Technologies, Carlsbad, CA, USA) on Ion Torrent PGM platform, contained 6 *BRCA1* and 5 *BRCA2* pathogenic variants previously identified (Table 1) plus 37 benign variants known (Table S1). Data obtained by Ion Torrent PGM sequencing were blindly analyzed by the Torrent Suite Software version 3.6 using the Variant Caller plugin (germline low stringency) and subsequently the specific json file. Straightforward default analysis only identified six out of 11 *BRCA1* and *BRCA2* pathogenic variants, four indel

**Table 3  *BRCA1* and *BRCA2* lower-performance and fwd/rev end to end unbalanced amplicons.**

| Gene | Id amplicon | GC content | Self-annealing | GC clamp | Hairpin formation |
|------|-------------|-----------|----------------|----------|-------------------|
| *BRCA1* | AMPL225438570 | fwd 43% | / | / | / |
|  |  | rev 36% | / | / | / |
| *BRCA2* | AMPL223392219 | fwd 41% | / | / | / |
|  |  | rev 27% | Yes | / | Yes |
| *BRCA2* | AMPL223379892 | fwd 22% | Yes | / | Yes |
|  |  | rev 36% | Yes | / | Yes |
| *BRCA2* | AMPL223390724 | fwd 41% | / | / | / |
|  |  | rev 39% | / | / | / |
| *BRCA2* | AMPL225505032 | fwd 46% | Yes | / | Yes |
|  |  | rev 36% | Yes | Yes | / |
| *BRCA2* | AMPL225504179 | fwd 33% | / | / | / |
|  |  | rev 36% | Yes | Yes | / |
| *BRCA1* | AMPL223735053 | fwd 38% | / | / | / |
|  |  | rev 48% | / | / | / |
| *BRCA1* | AMPL223413081 | fwd 46% | / | / | / |
|  |  | rev 28% | Yes | / | Yes |
| *BRCA1* | AMPL223712774 | fwd 48% | / | / | / |
|  |  | rev 39% | Yes | / | Yes |
| *BRCA2* | AMPL224626553 | fwd 24% | Yes | / | Yes |
|  |  | rev 50% | / | / | / |
| *BRCA2* | AMPL223938117 | fwd 31% | / | Yes | / |
|  |  | rev 39% | Yes | / | Yes |
| *BRCA2* | AMPL223512592 | fwd 25% | Yes | / | Yes |
|  |  | rev 46% | / | / | / |
| *BRCA2* | AMPL225349438 | fwd 38% | Yes | / | Yes |
|  |  | rev 42% | Yes | / | / |
| *BRCA2* | AMPL223730984 | fwd 50% | / | / | / |
|  |  | rev 41% | / | / | / |
| *BRCA2* | AMPL223515418 | fw 39% | Yes | / | Yes |
|  |  | rev 55% | Yes | / | / |
| *BRCA2* | AMPL225441321 | fw 52% | / | / | / |
|  |  | rev 48% | Yes | / | Yes |
| *BRCA2* | AMPL223959719 | fw 38% | / | / | / |
|  |  | rev 41% | Yes | / | Yes |

and two SNV (Fig. 2; Fig. S1). Setting the kmer length parameter value to 11 instead of the default value 19, allowed detection of the otherwise missed *BRCA1* indel c.4964_4982del19 (p.Ser1655Tyrfs) localized at the end of exon 16 (Fig. 3A), without generating FN.

To detect the *BRCA2* complex variant c.7921_7926delGAATTTinsAG (p.Glu2641Argfs) the generation of "complex variant candidates parameter" in the Torrent Suite Software has been activated (Fig. 3B). In this way, the NGS analysis correctly identified a total
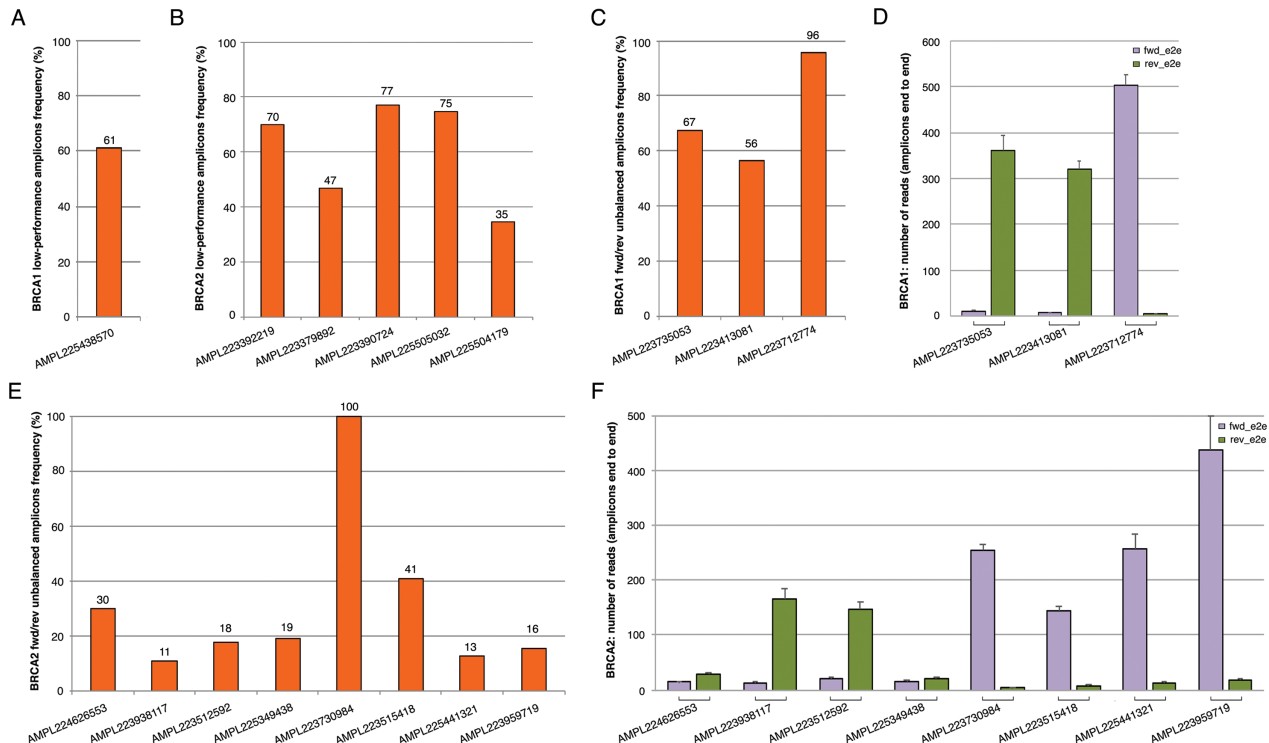

**Figure 1 Coverage plots of *BRCA1* and *BRCA2* lower-performance amplicons from 147 samples (TS and VS samples).** (A, B) *BRCA1* and *BRCA2* amplicons that did not reach a minimum coverage of 100×; (C, E) percentage of *BRCA1* and *BRCA2* fwd/rev unbalanced amplicons (minimum average depth in strand bias of 30× for each strand end to end); (D, F) histogram of the mean read depth per fwd/rev end to end unbalanced amplicons in *BRCA1* and *BRCA2*. Error bars represent +/- the standard error (SE) of reads number.

of 179 variants (68 in *BRCA1* and 111 in *BRCA2*), reaching the maximum sensitivity of 98.3% (95.3% for *BRCA1* and 100% for *BRCA2*) identifying eight out 11 pathogenic variants, missing only the three *BRCA1* LGRs (Table 4). NGS called eight FPs variants (seven in BRCA1 and one in BRCA2) in the 11 samples of TS in the homopolymeric regions of the genes. The positive predictive value, calculated as (TPs/TPs+ FPs), was 95.5% (89.7% for *BRCA1* and 99.1% for *BRCA2*). The panel covers 16,246 nucleotides (5,989 for *BRCA1* and 10,257 for *BRCA2*) for patient, then we covered 178,706 nucleotides (65,879 for *BRCA1* and 112,827 for *BRCA2*) in the TS. If we consider that TNs could be calculated as the total analyzed nucleotides-TPs-FPs-FNs, we could calculate a total of 178,524 TNs, that is 65,808 for *BRCA1* and 112,716 for *BRCA2*. The specificity, calculated as TN/(TN+FP), and the NPV, calculated as TN/(TN+FN), were 100% (Table 4). Therefore, considering LGRs as FNs, the concordance with Sanger sequencing analysis was 100%.

All these data indicated that an appropriate setting of the analytical pipeline can affect the accuracy of a variant call.

This optimized pipeline was then blindly applied on the VS of 136 samples, whose genotype was unknown. We detected pathogenic variants in 30 (22%) cases, 20 in *BRCA1* (15%), 10 in *BRCA2* (7%), 1 *BRCA1* VUS, and 1 *BRCA2* VUS. All variants were
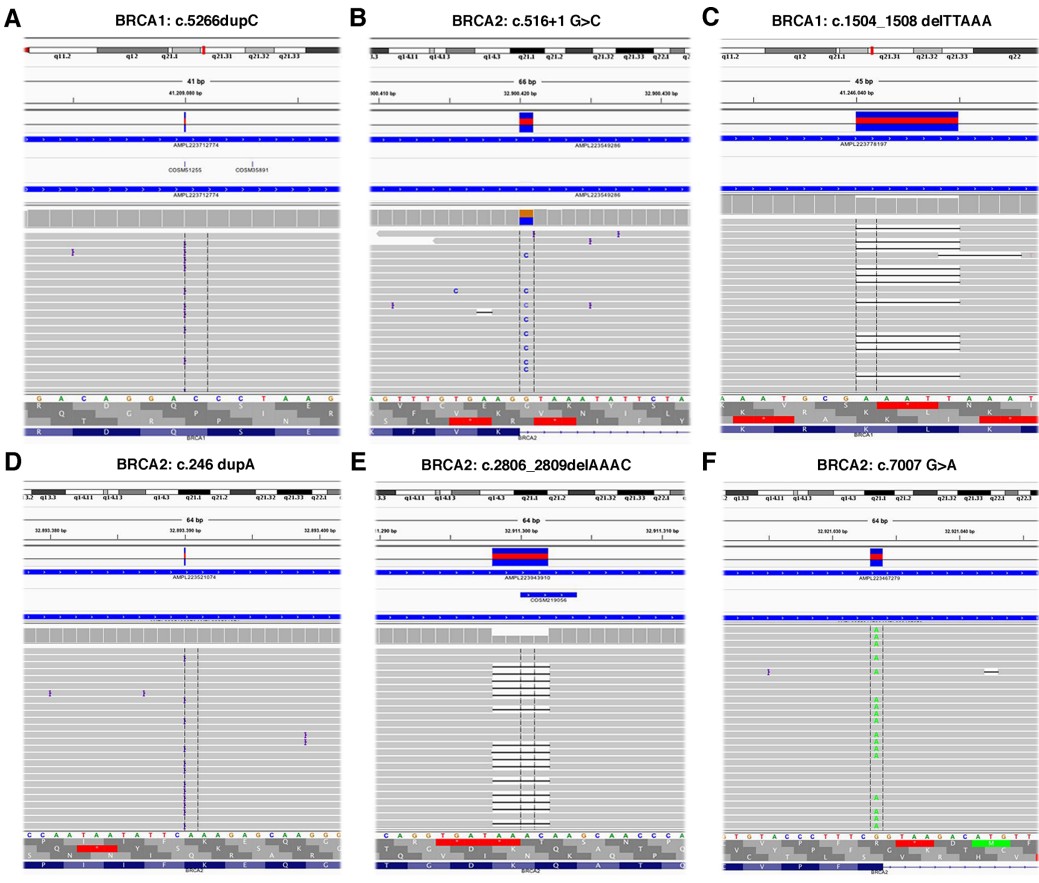

**Figure 2 Representative image of read alignments visualized with IGV showing the germline mutations detected by NGS default analysis (A–F).**

confirmed by Sanger sequencing in their correct zygosity status (Table 5). The remain 104 samples, negative for pathogenic variants, subjected to MLPA analysis were also negative for the presence of LGRs. Of note, one of the *BRCA2* deleterious mutations (c.72delA) was a novel disease-causing variant at exon 3, predicted to code for an early truncated protein (p.Gly25Aspfs), thus falling into class V (*Plon et al., 2008*; *ENIGMA consortium*: https://enigmaconsortium.org/library/general-documents/enigma-classification-criteria/). The novel mutation has been identified in one HBC family having a clear dominant inheritance pattern and with three cases of breast cancer before 40 years of age (Fig. S2). It has not yet been possible to perform segregation analysis in this family.

A complete *BRCA1* and *BRCA2* Sanger sequencing has been performed in 60 negative samples of VS (included in the 104 negative samples), because their higher expectation to be carrier of pathogenic mutations. As expected, Sanger sequencing not only confirmed the absence of *BRCA1* and *BRCA2* pathogenic variants, but also identified all benign variants in their correct zygosity status as detected by NGS (Table S4). The variants located in regions not covered by the respective primers have been excluded from this evaluation. Altogether, our data strongly suggest that Ion Torrent PGM and Ion AmpliSeq BRCA1 and BRCA2 Panel (Life Technologies, Carlsbad, CA, USA) represent

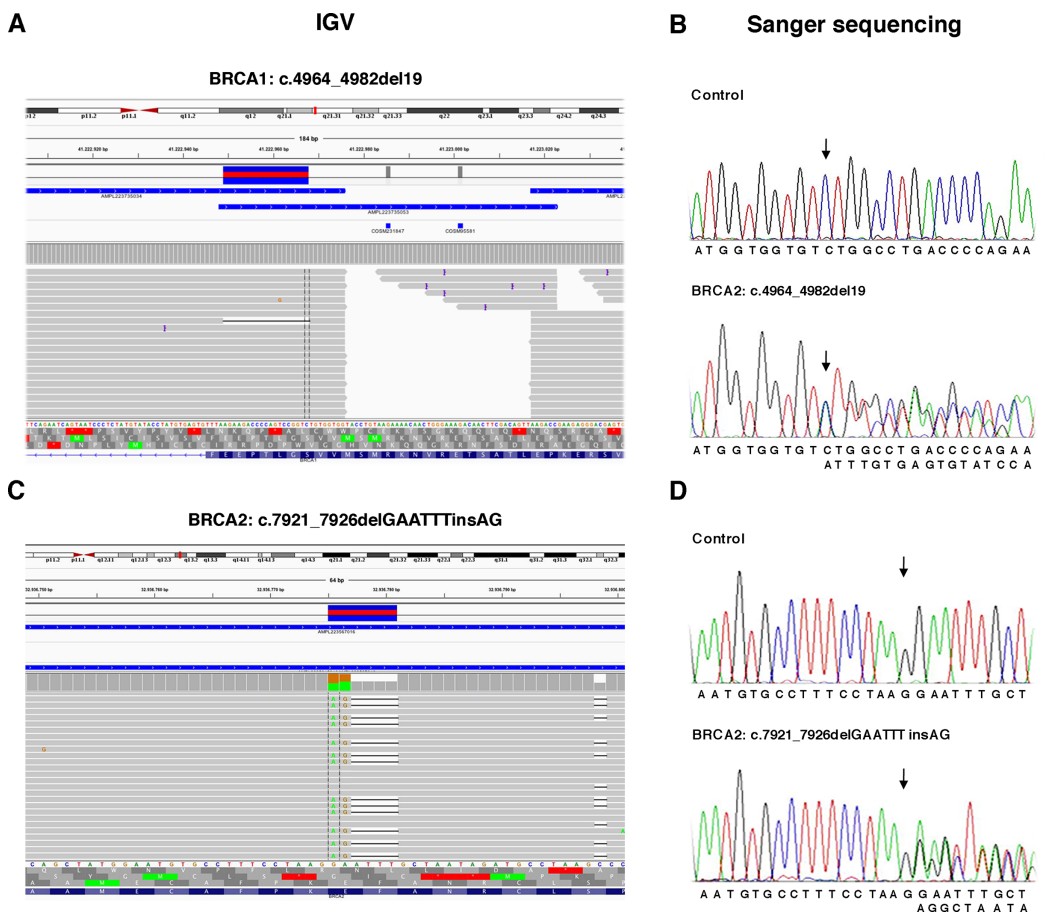

**Figure 3 Representative image of read alignments showing the germline mutations of *BRCA1* and *BRCA2* detected by NGS analysis after minimal adjustments in the analytical pipeline of Torrent Suite Software.** Read alignments visualized with IGV showing the germline mutation of *BRCA1*, c.4964_4982del19 (p.Ser1655Tyrfs) and *BRCA2*, c.7921_7926delGAATTTinsAG (p.Glu2641Argfs) (A and C). On the right side of the image, representative validation of NGS data by Sanger sequencing. Germline mutations were detected in the patients but not in the control subject. Arrows indicate the position of the mutated nucleotides (B and D).               

an excellent system to be applied in the field of diagnostics with 100% of concordance and sensitivity compared to Sanger method.

Our study enabled us to generate a standardized diagnostic workflow useful in clinical practice to quickly screen the *BRCA1* and *BRCA2* genes (Fig. 4).

## Cost efficiency and turn-around time

In comparison to traditional Sanger-based sequencing, the NGS workflow led to a reduction in cost, analysis time, and TAT for the screening of *BRCA1* and *BRCA2* genes. Considering the routine of our laboratory, which include PCR and Sanger sequencing, we calculated that we analyzed eight samples in 30–40 days. While, with the advent of NGS methodology we can analyze eight patients in 10 days, making the results available for the clinician in 15 working days following blood sampling. Costs entailed are obviously site-dependent, we calculated a cost of about 1,500–3,000 euro/sample for *BRCA1* and

**Table 4 Performance of the NGS assay (NGS *vs* Sanger) evaluated in the training set (in brackets, values were obtained considering LGRs as FNs).**

|  | BRCA1 | BRCA2 | BRCA1 + BRCA2 |
| --- | --- | --- | --- |
| Total variants | 68 | 111 | 179 |
| Called positions per patient | 5,989 | 10,257 | 16,246 |
| Total called positions | 65,879 | 112,827 | 178,706 |
| TP | 61 | 110 | 171 |
| FP | 7 | 1 | 8 |
| TN | 65,808 | 112,716 | 178,524 |
| FN | 0 (3) | 0 | 0 (3) |
| Sensitivity, TP/(TP + FN) | 100% (95.3%) | 100% | 100% (98.3%) |
| Specificity, TN/(TN + FP) | 100% | 100% | 100% |
| Positive predictive value (PPV), TP/(TP + FP) | 89.7% | 99.1% | 95.5% |
| Negative predictive value (NPV), TN/(TN + FN) | 100% | 100% | 100% |

*BRCA2* Sanger-based sequencing, and a consistent reduction to about 300 euro/sample with NGS.

## DISCUSSION

Next-generation sequencing methods are considered a promising technology, that reduces the cost and TAT of sequencing analysis. It can also be used to analyze multiple genes in a single run, greatly simplifying the study of complex hereditary disease where Sanger sequencing is not technically or economically feasible. Hence, many molecular genetic laboratories, including ours, are starting to use routinely this technology for diagnostic purpose (*De Leeneer et al., 2011*; *Michils et al., 2012*; *Chan et al., 2012*; *Feliubadaló et al., 2013*).

Recently, different platforms have been validated showing a comparable quality of data and throughput capacity (*Voelkerding, Dames & Durtschi, 2009*; *Rothberg et al., 2011*; *Loman et al., 2012*; *Wallace, 2016*). In this study, we validated the Ion Torrent PGM platform in combination with the ready-to-use Ion AmpliSeq BRCA1 and BRCA2 Panel (Life Technologies, Carlsbad, CA, USA), as, in our opinion, it may be a good candidate for clinical *BRCA1* and *BRCA2* sequencing. *BRCA* genes are also involved in the development of sporadic breast and ovarian tumors; and patients, carrying a germline or somatic *BRCA1/BRCA2* variants, may benefit from PARP inhibitors therapy (*Coleman et al., 2017*; *Robson et al., 2017*; *Oza et al., 2018*; *Litton et al., 2018*). Investigating *BRCA1* and *BRCA2* mutation status in ovarian and breast cancer tissues patients, has a key role in addressing the therapeutic choices. Several studies demonstrated that NGS could be an efficient method for detecting somatic mutations in *BRCA1* and *BRCA2* genes using DNA from formalin-fixed and paraffin-embedded (FFPE) breast and ovarian cancer tissues (*Ellison et al., 2015*; *Mafficini et al., 2016*; *Enyedi et al., 2016*). In our laboratory the Ion Torrent PGM platform is currently used for the routine molecular characterization of FFPE tumor tissues from colon or lung cancers patients (*Belardinilli et al., 2015*) with a highly satisfactory performance.

**Table 5 BRCA1 and BRCA2 variants contained in the validation set.**

| Variant | | Gene | Effect | Type | Number of times a variant was observed |
|---|---|---|---|---|---|
| DNA | Protein | | | | |
| c.1360_1361delAG | p.Ser454Terfs | BRCA1 | Pathogenic | del | 1 |
| c.5062_5064delGTT | p.Val1688del | BRCA1 | Pathogenic | del | 1 |
| c.5266dupC | p.Gln1756Profs | BRCA1 | Pathogenic | dup | 3 |
| c.1462dupA | p.Thr488Asnfs | BRCA1 | Pathogenic | dup | 1 |
| c.1016dupA | p.Val340Glyfs | BRCA1 | Pathogenic | dup | 1 |
| c.3331_3334delCAAG | p.Glu1111Asnfs | BRCA1 | Pathogenic | del | 1 |
| c.4117G>T | p.Glu1373Ter | BRCA1 | Pathogenic | SNV | 6 |
| c.1505T>G | p.Leu502Ter | BRCA1 | Pathogenic | SNV | 1 |
| c.5074+1G>T | p.? | BRCA1 | Pathogenic | SNV | 1 |
| c.181T>G | p.Cys61Gly | BRCA1 | Pathogenic | SNV | 1 |
| c.1016dupA | p.Val340Glyfs | BRCA1 | Pathogenic | dup | 1 |
| c.5335_5335delC | p.Gln1779Asnfs | BRCA1 | Pathogenic | del | 1 |
| c.1881C>G | (p.Val627=) | BRCA1 | Uncertain significance | SNV | 1 |
| c.5123C>A | p.Ala1708Glu | BRCA1 | Pathogenic | SNV | 1 |
| c.9252_9255delAACAinsTT | p.Lys3084Asnfs | BRCA2 | Pathogenic | Complex | 1 |
| c.4131_4132insTGAGGA | p.Thr1378_Gly1712delinsTer | BRCA2 | Pathogenic | ins | 1 |
| c.1238_1238delT | p.Leu413Hisfs | BRCA2 | Pathogenic | del | 1 |
| c.5211_5214delTACT | p.Asp1737_1738delThrfs | BRCA2 | Pathogenic | del | 1 |
| c.5718_5719delCT | p.Leu1908Argfs | BRCA2 | Pathogenic | del | 1 |
| c 2684_2684delC | p.Ala895Valfs | BRCA2 | Pathogenic | del | 1 |
| c.6591_6592delTG | p.Glu2198Asnfs | BRCA2 | Pathogenic | del | 1 |
| c.7007G>A | p.Arg2336His | BRCA2 | Pathogenic | SNV | 1 |
| c.7008-62A>G | p.? | BRCA2 | Uncertain significance | SNV | 1 |
| c.72delA | p.Gly25Aspfs | BRCA2 | Pathogenic | del | 1 |
| c.6275_6276delTT | p.Leu2092Profs | BRCA2 | Pathogenic | del | 1 |

Note:
Abbreviations: del, deletion; dup, duplication; ins, insertion; SNV, single nucleotide variant.

The accuracy of NGS approach represents an important requirement in diagnostic applications, and it strongly depends on the read depth and on the analytical pipeline (*Goldfeder et al., 2016*). In our tests, we achieved an acceptable sequencing depth and average uniformity of amplicons coverage in all samples. 60 of the 104 *BRCA1* and *BRCA2* negative samples, that showed a higher expectation to be carrier of pathogenic mutations (i.e. HBC with BRCAPro5.1 greater than 40%, Male Breast Cancer or HBOC cases), were also subjected to *BRCA1* and *BRCA2* Sanger sequencing. We observed 100% sensitivity and 100% concordance as well as correct estimate heterozygote and homozygote status.

A limit of Ion AmpliSeq BRCA1 and BRCA2 Panel (Life Technologies, Carlsbad, CA, USA) is represented by the small group of lower-performance amplicons, that either

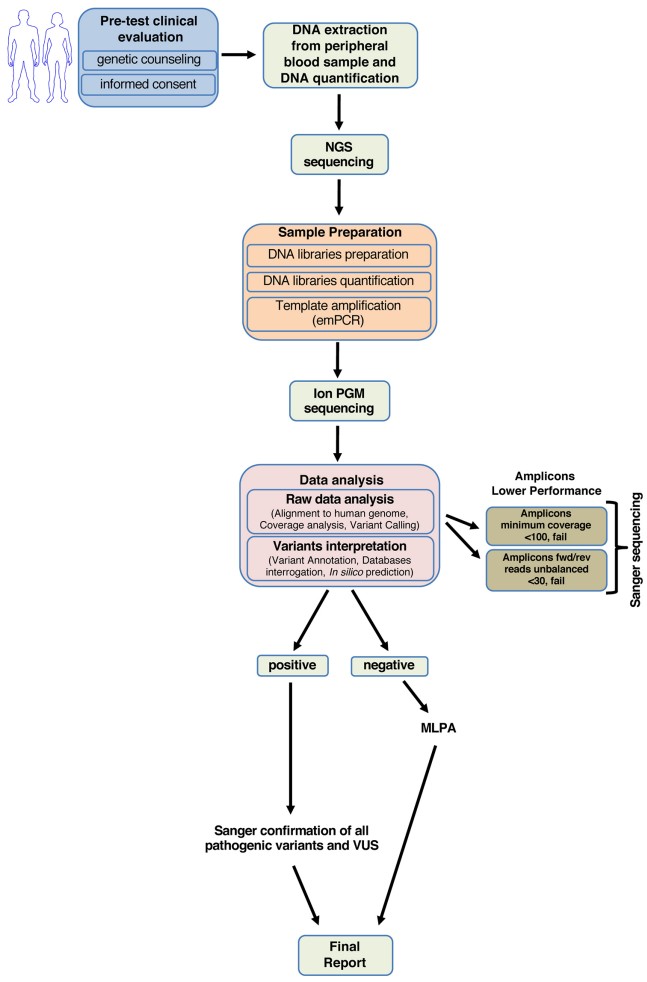

**Figure 4 Workflow for analyzing _BRCA1_ and _BRCA2_ using NGS.**

did not reach a minimum coverage of 100×, as also reported by _Pilato et al. (2016)_, or showed fwd/rev end to end unbalance. The mean amplicon read depth could be affected by the large A-T reach region or GC content, self-annealing/hairpin formation and/or GC clamp issues (_Yan et al., 2016_). We overcame these limits performing, in all samples, Sanger sequencing for the regions covered by low-performance amplicons. In addition, the presence of highly homopolymeric regions in the _BRCA1_ and _BRCA2_ genes and the nature of semiconductor sequencing chemistry of Ion Torrent PGM were shown to limit the correct call of indel variants in these regions (_Yeo et al., 2014_; _Zanella et al., 2017_). Nonetheless, we did not observe incorrect calls in these regions in our samples.

A complete clinical level analysis of _BRCA1_ and _BRCA2_ includes the study of LGRs. In our experience, the Ion AmpliSeq BRCA1 and BRCA2 Panel (Life Technologies, Carlsbad, CA, USA) was not feasible for analyzing copy number alteration due to the sequencing read length (range, 71–239 bp) and to the high number of FP obtained (_Pilato et al., 2016_). Therefore, the MLPA analysis is always required using this panel.

More recently, Thermo Fisher Scientific's developed Oncomine® BRCA Research Panel, that seems to solve all these issues (*Hirotsu et al., 2017*).

Noteworthy, we identified also a novel disease-causing mutation at exon 3 of *BRCA2* gene in one HBC family. Our results demonstrated that Torrent Suite Software with the appropriate adjustments achieved 100% sensitivity (considering LGRs as FNs) and the 100% concordance with Sanger sequencing analysis, as well as correct estimate heterozygote and homozygote status from variant frequencies.

## CONCLUSIONS

In conclusion, our study, according to other published works (*Feliubadaló et al., 2013*; *Yeo et al., 2014*; *Trujillano et al., 2015*), shows that NGS performed with a commercial panel (Ion AmpliSeq BRCA1 and BRCA2 Panel; Life Technologies, Carlsbad, CA, USA) is highly efficient for the detection of germline mutations in *BRCA1* and *BRCA2* genes using DNA samples from routinely available patients. Moreover, the costs of NGS technology are largely lower (about 300 Euros/patient), compared with conventional methods, with a substantial reduction (about 50%) in analysis time and TAT. In our experience, the NGS achieves a perfect sensitivity and the accuracy of Sanger sequencing, becoming an indispensable instrument for routine diagnostic testing on *BRCA1* and *BRCA2* genes.

## ACKNOWLEDGEMENTS

The authors would like to thank Valentina Silvestri for advice on data analysis.

### Funding

This work was supported by grants from the Associazione Italiana per la Ricerca sul Cancro (IG17734 to Giuseppe Giannini; IG16933 to Laura Ottini); Ministry of University and Research, PRIN projects (Giuseppe Giannini); Istituto Pasteur-Fondazione Cenci Bolognetti (Giuseppe Giannini). Virginia Valentini is the recipient of a fellowship of the PhD Programme in Tecnologie Biomediche in Medicina Clinica, University La Sapienza. Francesca Belardinilli is the recipient of a fellowship from Fondazione Umberto Veronesi. The funders had no role in study design, data collection and analysis, decision to publish, or preparation of the manuscript.

### Grant Disclosures

The following grant information was disclosed by the authors:
Associazione Italiana per la Ricerca sul Cancro: IG17734, IG16933.
Ministry of University and Research, PRIN projects.
Istituto Pasteur-Fondazione Cenci Bolognetti.
Tecnologie Biomediche in Medicina Clinica, University La Sapienza.
Fondazione Umberto Veronesi.

### Competing Interests

The authors declare that they have no competing interests.

## Author Contributions

- Arianna Nicolussi conceived and designed the experiments, performed the experiments, analyzed the data, prepared figures and/or tables, approved the final draft.
- Francesca Belardinilli performed the experiments, approved the final draft.
- Yasaman Mahdavian performed the experiments, approved the final draft.
- Valeria Colicchia contributed reagents/materials/analysis tools, approved the final draft.
- Sonia D'Inzeo performed the experiments, prepared figures and/or tables, approved the final draft.
- Marialaura Petroni contributed reagents/materials/analysis tools, approved the final draft.
- Massimo Zani contributed reagents/materials/analysis tools, approved the final draft.
- Sergio Ferraro contributed reagents/materials/analysis tools, approved the final draft.
- Virginia Valentini performed the experiments, approved the final draft.
- Laura Ottini analyzed the data, authored or reviewed drafts of the paper, approved the final draft.
- Giuseppe Giannini conceived and designed the experiments, authored or reviewed drafts of the paper, approved the final draft.
- Carlo Capalbo conceived and designed the experiments, analyzed the data, authored or reviewed drafts of the paper, approved the final draft.
- Anna Coppa conceived and designed the experiments, analyzed the data, authored or reviewed drafts of the paper, approved the final draft.

## Human Ethics

The following information was supplied relating to ethical approvals (i.e., approving body and any reference numbers):

The University Sapienza of Rome granted Ethical approval to carry out the study. (Rif.Ce:4903; Prot. n. 88/18).

## DNA Deposition

The following information was supplied regarding the deposition of DNA sequences:

The DNA sequence is available in the ClinVar database: SCV000854769.

## Data Availability

The raw data is available in the Supplemental Information.

## Supplemental Information

Supplemental information for this article can be found online at http://dx.doi.org/10.7717/peerj.6661#supplemental-information.

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
