# Peer review of "Next-generation sequencing of BRCA1 and BRCA2 genes for rapid detection of germline mutations in hereditary breast/ovarian cancer"

_PeerJ, doi:10.7717/peerj.6661_

## Round 0.1 · original submission · Major Revisions

Please revise your manuscript in line with the reviewer comments.

Reviewer 1 ·

Basic reporting

See below

Experimental design

See below

Validity of the findings

See below

Additional comments

Thank you for asking me to review the manuscript by Nicolussi et al. entitled: “Next-Generation Sequencing of BRCA1 and BRCA2 genes for rapid 2 detection of germline mutations in Hereditary Breast/Ovarian Cancer”

The study describes the use of Ion Torrent (semiconductor sequencing) technology to sequence germline DNA for BRCA1/2 variants. This providers of this technology claim that it provides a cheaper and faster method of detecting variants.
Nicolussi et al. show that the technology can be used to detect germline BRCA1/2 variants in the training set (n=11) and then validate the technology on a large series (n=136 probands), with additional Sanger sequencing of 60 samples as a gold-standard sequencing techniques.
The authors show for SNV/indels detection concordance with Sanger sequencing of 100% (excluding LGRs). The authors also detect 30 BRCA1/2 pathogenic variants in the VS, and confirmed these variants using Sanger sequencing.

Major points:
• Introduction: The introduction needs more explanation of the Ion Torrent PGM beyond… “The Ion Torrent PGM is a new sequencing platform that substantially differs from other sequencing technologies by measuring pH rather than light to detect polymerization events.”
• Results: Could the authors clarify if they detected any pathogenic BRCA1/2 variants using Sanger sequencing that were not detected using the Ion Torrent technology please? I.e. False-negative rate.
• Results: Are the authors able to provide the sensitivity, specificity, PPV and NPV for SNV/indels in a table format with Sanger Sequencing as the gold standard?
• Results: There was no information/discussion regarding time the assay took and costs, until it is briefly mentioned in the conclusion.
• Discussion: The discussion needs more emphasis on the broader implications of using the technology within a laboratory e.g. why should we change our current platforms to use this technology?
• Discussion: The general conclusion of the paper is that Ion can be used to detect SNV, indels with excellent sensitivity, but fails to detect LGR. Therefore MLPA should be continued to be used as a method for detecting LGRs. This conclusion needs clearly stating.

Minor points:
1. The following reference should be included regarding the risk of breast/ovarian cancer in BRCA1/2 carriers – Kuchenbaecker et al. JAMA 2017
2. The introduction requires a sentence regarding the clinical utility of germline BRCA1/2 testing for breast and ovarian cancer therapies (please see following clinical trials: SOLO1, SOLO2, ARIEL3, NOVA, EMBRACA, OLYMPIAD – all phase 3 PARPi trials).
3. Is studio the correct terminology in this statement? “The BRCA1 and BRCA2 panel used in this studio generates 167 amplicons that cover all targeted coding exons and exon-intron boundaries”
4. A flowchart for the methodology would be nice as the description is very detailed
5. In the discussion, could the authors make a comment on whether they feel this technology can be applied to DNA extracted from breast/ovarian cancer tumour tissue please?
6. In Table 1, please change the LGR DNA level to describe the Exon deleted instead (e.g. Exon 13 duplication).
7. In Table 1 and 4, the “reference ID according to NCBI” can be removed as this is irrelevant
8. In Tables 1 and 4, please define what the column “quality” means?
9. In Table 4, a number of the variants are defined as indels, but they are actually small deletions or duplications, not insertion-deletions
10. In Table 4, the variant c.4131_4132insTGAGGA is not “complex” – it is an “insertion”
11. In Figure 4, where any other family members tested for this variant in germline/tumour DNA as part of segregation analysis?

Reviewer 2 ·

Basic reporting

The manuscript includes a relevant topic and introduces the rationale for the project. Overall the manuscript reads well and follows the standard manuscript structure. A more detailed legend for Figure 1 D en F would be helpful, especially as the have the appearance of a boxplot while they depict other type of information with a SD. A minor detail is the varying use of the terms mutations and (pathogenic) variants, which may be confusing.

Experimental design

-The aim of the study could be clarified. The aim stated in abstract and in the introduction seems to differ slightly. The final sentences of the introduction read as a conclusion rather than a study aim.
-The method section provides structured information of the different analyses steps. The outcomes measurements of the comparison of both methods are not included. Please include a short statements on this.
-The distinction between the training set and validation set is clear. Though it would be useful to clearify the rationale for the different number of patients included in various analyses (e.g. subset of 11 TS + 60 VS samples for Sanger sequencing [methods]; a critical analyses of coverage in 73 VS sample [results]).

Validity of the findings

-The detection of all types of variant is evaluated between the NGS method and the conventional methods (sanger/MLPA), and therefore the training set includes SNVs, indels and large genomic regions. However, the methods and conclusion of the comparison are based on the SNVs or indels detected by NGS method versus Sanger sequencing. This should be clarified in the aim and methods of the manuscript, and the impact of missing the LGRs should be addressed in the results and discussion section.
-The conclusion of the study includes statements on the performance regarding costs and turn around times, which has not been addressed in the main text as outcome measurements and data.
-The validation set consists of 136 samples, of which 30 with a pathogenic variant, 2 VUS, and 60 no pathogenic variant. What were the findings for the other 44 samples? In case these where cases without pathogenic variants and a less extensive personal and family history of cancer, please discuss why for this group no additional Sanger sequencing has been performed as BRCA1/2 variants are present in patients with less suspect phenotypes.

Additional comments

The authors address an important topic. Given the increasing demand for genetic diagnostics, improvement in the efficient use of means and time, meanwhile ensuring quality, is a pressing issue.

·

Basic reporting

1. Well structured paper with clearly written methods, results and discussion.

2. Whilst the novel variant you described (c.72delA) is an interesting finding to report, I did not find that it was completely necessary to include the pedigree (Fig 4) in your main manuscript. Could this be included in supplementary material? I would also be interested to know if the variant segregated with the affected family members in Family BR1270.

3. Could you include criteria used to enrol the probands for BRCA1 or BRCA2 sequencing in either the main manuscript or supplementary material.

4. Last paragraph of Results section, lines 243 to 252: Some elements of this paragraph may be better suited in the Discussion section where you are interpreting the significance of your findings in light of your research. You could therefore make the last section of your results more concise and fact-based.

5. Your discussion states that the sequencing technology has 100% sensitivity if large genomic rearrangements are counted as false negatives. Are there any other methods you could undertake or suggestions that you have to improve the detection of LGRs?

6. Please check references for spelling errors.

Experimental design

1. You have mentioned the guidelines for variant interpretation from Plon et al, 2008 in the analysis of the novel variant. We use American College of Medical Genetics guidelines for variant interpretation (2015) in our local centre and I therefore wondered why you were not using more updated guidelines for your variant analysis?

Validity of the findings

No comment

---

## Round 0.2 · accepted · Accept

Congratulations you have addressed all comments well. Please just add the legends as requested

# Reviewer 1 ·

Basic reporting

Please see below.

Experimental design

Please see below.

Validity of the findings

Please see below.

Additional comments

The authors have addressed all points raised, thank you.

Reviewer 2 ·

Basic reporting

The authors have addressed the previous remarks.
The supplemental files do not include any table/figure number, title and description. Please add these.

Experimental design

No comment

Validity of the findings

No comment